# TRIM59: A potential diagnostic and prognostic biomarker in human tumors

**Zheng Jin[1], Liping Liu[1], Youran Yu[2], Dong Li[1], Xun Zhu[1], Dongmei Yan[1☯*],**
**Zhenhua Zhu** (iD) **[3☯*]**

**1** Department of Immunology, College of Basic Medical Sciences, Jilin University, Changchun City, Jilin Province, China, **2** College of Pharmacy, Shanghai University of Medicine & Health Sciences, Shanghai, China, **3** Department of Orthopaedic Trauma, Center for Orthopaedic Surgery, The Third Affiliated Hospital of Southern Medical University, Guangzhou, Guangdong Province, China

☯ These authors contributed equally to this work.
* dmyan@jlu.edu.cn (DY); zhuzhenhua93@hotmail.com (ZZ)

## Abstract

TRIM59 is a protein that is highly expressed in a variety of tumors and promotes tumor development. However, the use of TRIM59 as tumor diagnosis and prognosis biomarker has not been fully explored. We collected datasets from the cancer genome atlas (TCGA) and gene expression omnibus (GEO) to investigate its potential as a biomarker for diagnosis and prognosis. A total of 46 studies, including 11,558 patients were included in this study. Here, we showed that TRIM59 was significantly upregulated in 15 type of human solid tumors in comparison to their adjacent tissues. Receiver operating characteristic curve (ROC) results provided further evidence for the use of TRIM59 as a potential tumor diagnosis biomarker. Overall survival (OS) was compared between TRIM59 high expression and low expression groups. High expression of TRIM59 indicated a poor prognosis in multiple solid tumors. Taken together, these analyses showed that TRIM59 was upregulated in various types of tumors and had the potential to be used as a diagnostic and prognostic biomarker in human solid tumors.

## Introduction

Cancer has always been a major global health concern and creates a heavy financial burden for patients and the health care system [1]. It is estimated that there might be 18.1 million new cancer cases and 9.6 million cancer deaths in 2018 [2]. Cancers are always characterized by a group of abnormal expressed genes, and these genes have the potential to be used as diagnosis markers or prognosis predictors [3, 4].

TRIM59 is a member of TRIM protein family, which is comprised of a RING domain, b-box domain, coiled helix domain, and c-terminal non-specific domain [5]. Recent studies have shown that TRIM59 played a role in epigenetic modification [6], embryonic development [7] and autophagy [8], and was upregulated in multiple tumors [9]. It has been found that down-regulation of TRIM59 significantly inhibited tumor growth in *vitro* experiments and animal models [10, 11]. However, no large-scale clinical studies have been performed to demonstrate the relationship between TRIM59 and tumor diagnosis and prognosis.

GSE42568, GSE48390, GSE58812, GSE102287, GSE29013, GSE30219, GSE31210, GSE3141, GSE19188, GSE37745, GSE50081, GSE26193, GSE32062, GSE63885, GSE18520, GSE15459, GSE57303, GSE62254, GSE17891, GSE17679, GSE19750, GSE108474, GSE7696); https://xenabrowser.net (Projects: LUAD, BRCA, UCEC, LUSC, HNSC, KIRC, PRAD, BLCA, THCA, KIRP, LIHC, STAD, COAD, READ, CHOL, CESC, ESCA, GBM, LGG, OV, PAAD, SARC, SKCM)

**Funding:** This work was supported by the National Natural Science Foundation of China (No. 81871245) and Department of Education of Jilin Province (JJKH20190095KJ).

**Competing interests:** The authors have declared that no competing interests exist.

In this study, the cancer genome atlas (TCGA) and gene expression omnibus (GEO) datasets were used to analyze the expression of TRIM59 in tumors and evaluate the value of TRIM59 as a biomarker for tumor diagnosis and prognosis.

# Materials and methods

## Data collection and extraction

Datasets containing tumor samples and corresponding normal samples were downloaded from TCGA. Microarrays containing overall survival (OS) information were searched in the GEO database. Inclusion criteria were as follows: (1) should be human tumor samples and each dataset should contain at least 30 independent samples; (2) for the convenience of data processing, only microarray data performed on GPL570 platform was retained; (3) characteristics of studies and survival information were reported or could be determined. Exclusion criteria include: repetitive studies containing same datasets or patient cohorts. A total of 26 GEO datasets were included in the study, including 6 breast cancer datasets (GSE20685, GSE20711, GSE16446, GSE42568, GSE48390, GSE58812), 8 lung cancer datasets (GSE102287, GSE29013, GSE30219, GSE31210, GSE3141, GSE19188, GSE37745, GSE50081), 4 ovarian cancer datasets (GSE26193, GSE32062, GSE63885, GSE18520), 3 gastric cancer datasets (GSE15459, GSE57303, GSE62254), 1 pancreatic cancer dataset (GSE17891), 1 Ewing sarcoma dataset (GSE17679), 1 adrenocortical carcinoma dataset (GSE19750), 2 brain cancer dataset (GSE108474, GSE7696).

## Data processing

Level 3 RNAseq and associated clinical information of each TCGA solid tumor project (LUAD-lung adenocarcinoma, BRCA-breast carcinoma, UCEC-uterine corpus endometrial carcinoma, LUSC-lung squamous cell carcinoma, HNSC-head and neck squamous cell carcinoma, KIRC-kidney renal clear cell carcinoma, PRAD-prostate adenocarcinoma, BLCA-bladder carcinoma, THCA-thyroid carcinoma, KIRP-kidney renal papillary cell carcinoma, LIHC-liver hepatocellular carcinoma, STAD-stomach adenocarcinoma, COAD-colon adenocarcinoma, READ-rectum adenocarcinoma, CHOL-cholangiocarcinoma, CESC-cervical squamous cell carcinoma and endocervical adenocarcinoma, ESCA-esophageal carcinoma, GBM-glioblastoma multiforme, LGG-lower grade glioma, OV-ovarian serous cystadenocarcinoma, PAAD-pancreatic adenocarcinoma, SARC-sarcoma, SKCM-skin cutaneous melanoma) was downloaded from UCSC Xena (https://xenabrowser.net/). Downloaded FPKM (Fragments Per Kilobase of exon model per Million mapped fragments) gene expression data was log2 transformed for the convenience of comparison. Samples' expression and clinical information was matched, samples without complete survival or expression information were excluded. TRIM59 expression in each project was presented with boxplot. As for GEO datasets, RMA (Robust Multichip Average) normalized gene expression data of GPL570 platform was downloaded and matched with associated clinical information.

## ROC, AUC and SROC analyses

R package "OptimalCutpoints" was first used to determine the optimal cutpoint in each TCGA project, at which the sample types (tumor or normal) could be best distinguished. Then the sample types were predicted by the TRIM59 expression level according to the cut point of each project using the "Youden" prediction model [12]. ROC and associated 95% confidence intervals (CIs) of each project was also calculated using the "optimal.cutpoints" function of package "OptimalCutpoints". Module "midas" of Stata 15.0 (StataCorp LLC, USA) was used

for the meta-analysis of ROC curves of all the projects. Sensitivity and specificity of the meta-analysis were evaluated, and the summary of ROCs (SROC) was calculated. Deeks' funnel plot asymmetry test was used to investigate potential publication bias in SROC analysis.

### Overall survival analysis

Datasets regarding TRIM59 expression and clinical characteristics of human cancers were downloaded from TCGA and GEO databases. According to the TRIM59 expression level in each data set, the samples were divided into high expression and low expression groups in comparison to the median expression level. The "survival" package was used to calculate the hazard ratio (HR) and 95% CIs for the high-expression TRIM59 group versus the low-expression TRIM59 group.

### Statistical analysis

All the analyses were performed on R (Version 3.6.1) and Stata 15.0 (StataCorp LLC, USA). Normalized expression data were downloaded from TCGA or GEO datasets. Unpaired t test was used for comparison between groups. As for ROC analysis, R package "OptimalCutpoints" was first used to calculate the optimal cutpoint for tumor diagnosis, then ROC and 95% CIs were calculated for each project. Stata module "medias" was used for ROC meta-analysis. R package "survival" was used for survival analysis and package "meta" was used for survival meta-analysis.

## Results

### TRIM59 was highly expressed in human tumors

Previous studies have shown that TRIM59 was highly expressed in a variety of tumors and closely related to the occurrence and development of tumors. We analyzed 15 types of tumor datasets that included sufficient numbers of tumors and adjacent normal tissues in TCGA, confirming that TRIM59 was highly expressed in tumor samples in comparison to their adjacent tissues (Fig 1).

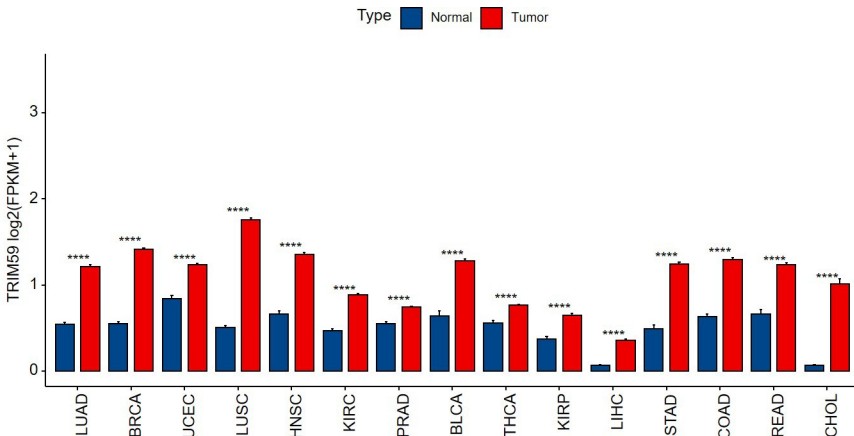

**Fig 1. Relative expression of TRIM59 in human tumors based on TCGA database, comparisons were conducted using unpaired t test.** ****p < 0.0001. FPKM (Fragments Per Kilobase of exon model per Million mapped fragments).

**Table 1. Characteristics of studies and AUC analyses.**

| ID | Projects | AUC (95% CI) | Normal Samples | Tumor Samples | Cut-off value | FP | FN | TP | TN |
|---|---|---|---|---|---|---|---|---|---|
| 1 | TCGA-LUAD | 0.928 (0.903, 0.952) | 59 | 526 | 0.768746 | 5 | 98 | 428 | 54 |
| 2 | TCGA-BRCA | 0.955 (0.94, 0.969) | 113 | 1104 | 0.882547 | 8 | 139 | 965 | 105 |
| 3 | TCGA-UCEC | 0.787 (0.737, 0.838) | 35 | 548 | 1.070402 | 3 | 195 | 353 | 32 |
| 4 | TCGA-LUSC | 0.985 (0.972, 0.998) | 49 | 501 | 0.846782 | 1 | 20 | 481 | 48 |
| 5 | TCGA-HNSC | 0.876 (0.837, 0.915) | 44 | 502 | 0.971722 | 3 | 137 | 365 | 41 |
| 6 | TCGA-KIRC | 0.912 (0.878, 0.946) | 72 | 535 | 0.573303 | 13 | 58 | 477 | 59 |
| 7 | TCGA-PRAD | 0.712 (0.644, 0.779) | 52 | 499 | 0.689123 | 10 | 235 | 264 | 42 |
| 8 | TCGA-BLCA | 0.905 (0.845, 0.965) | 19 | 411 | 0.828988 | 2 | 60 | 351 | 17 |
| 9 | TCGA-THCA | 0.742 (0.676, 0.809) | 58 | 510 | 0.690223 | 10 | 189 | 321 | 48 |
| 10 | TCGA-KIRP | 0.783 (0.706, 0.859) | 32 | 289 | 0.390211 | 11 | 62 | 227 | 21 |
| 11 | TCGA-LIHC | 0.921 (0.89, 0.952) | 50 | 374 | 0.104429 | 3 | 60 | 314 | 47 |
| 12 | TCGA-STAD | 0.951 (0.918, 0.984) | 32 | 375 | 0.691021 | 6 | 26 | 349 | 26 |
| 13 | TCGA-COAD | 0.941 (0.919, 0.963) | 41 | 471 | 0.943438 | 2 | 74 | 397 | 39 |
| 14 | TCGA-READ | 0.93 (0.889, 0.971) | 10 | 167 | 0.91301 | 0 | 24 | 143 | 10 |
| 15 | TCGA-CHOL | 1 (1, 1) | 9 | 36 | 0.287042 | 0 | 0 | 36 | 9 |

LUAD, lung adenocarcinoma; BRCA, breast carcinoma; UCEC, uterine corpus endometrial carcinoma; LUSC, lung squamous cell carcinoma; HNSC, head and neck squamous cell carcinoma; KIRC, kidney renal clear cell carcinoma; PRAD, prostate adenocarcinoma; BLCA, bladder carcinoma; THCA, thyroid carcinoma; KIRP, kidney renal papillary cell carcinoma; LIHC, liver hepatocellular carcinoma; STAD, stomach adenocarcinoma; COAD, colon adenocarcinoma; READ, rectum adenocarcinoma; CHOL, cholangiocarcinoma.

## High expression of TRIM59 showed high efficacy in the diagnosis of human tumors

To explore the value of high expression of TRIM59 in tumors, we tested if TRIM59 could be used to identify healthy and tumor samples. ROC analysis was performed based on the data obtained from TCGA. Results presented in Table 1 demonstrated that TRIM59 showed high

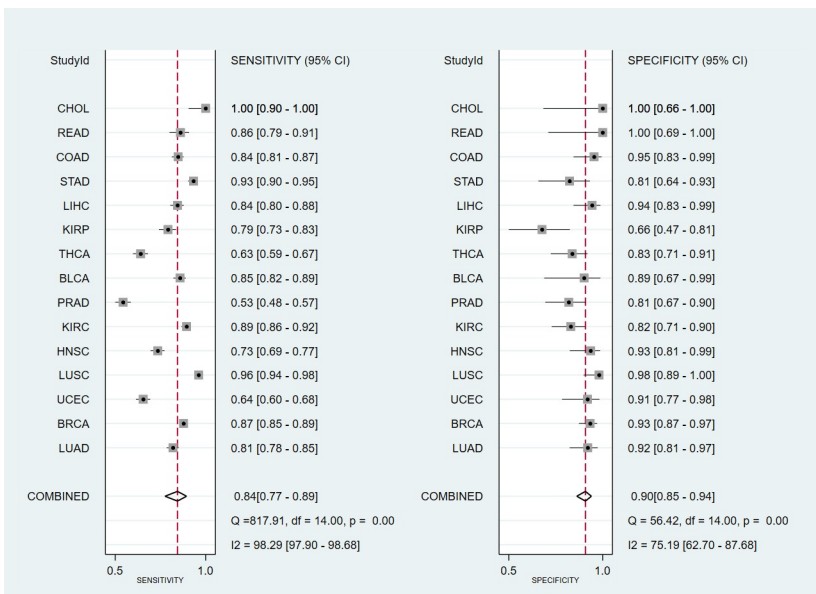

**Fig 2. Forest plot showing mean sensitivity and specificity with corresponding heterogeneity statistics for the prediction of sample types with the expression level of TRIM59.**

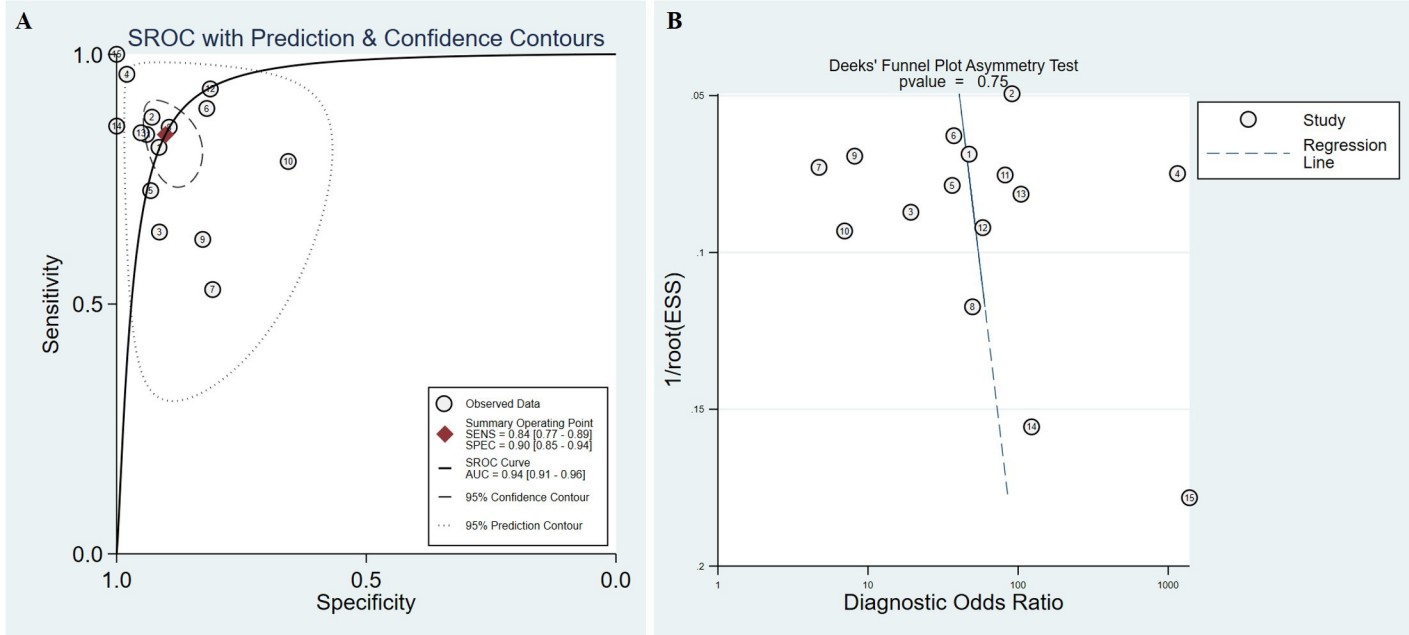

**Fig 3. Summary ROC analysis.** (A) Summary ROC for the evaluation of prediction efficacy of TRIM59, with confidence and prediction regions around mean operating sensitivity and specific point. The ID in the figure corresponds to the projects in Table 1. (B) Deeks' funnel plot asymmetry test showing the potential publication bias. The ID in the figure corresponds to the projects in Table 1.

diagnosis efficacy in multiple cancers, especially in CHOL, with a prediction accuracy up to 100%. The pooled sensitivity and specificity were 0.84 (95% CI:0.77–0.89) and 0.90 (95% CI:0.85–0.94) respectively (Fig 2). Additionally, the area under ROC (AUC) of the SROC was 0.94 (95% CI:0.91–0.96) (Fig 3A). In order to test the potential publication bias, Deeks' funnel plot asymmetry test was performed and revealed that no significant substantial publication bias was found in SROC analysis (p = 0.75) (Fig 3B). These results suggested that the high expression of TRIM59 could be used as a diagnostic marker in different types of human cancers.

## Expression of TRIM59 indicated prognosis in human tumors

To investigate the relationship between TRIM59 and tumor prognosis, OS meta-analysis was conducted. A total of 21 TCGA and 26 GEO datasets containing 11,558 patients were included (Table 2). High expression of TRIM59 indicated poor prognosis in KIRP (HR 3.18; 95%CI 1.76–5.75), LGG (HR 1.98; 95%CI 1.39–2.83), LUAD (HR 1.40; 95%CI 1.04–1.88), lung cancer (GSE30219: HR 1.44; 95%CI 1.09–1.89; GSE31210: HR 2.37; 95%CI 1.22–4.60), and indicated good prognosis in CESC (HR 0.56; 95%CI 0.34–0.92) and SKCM (HR 0.71; 95%CI 0.54–0.92) (Fig 4).

## Discussion

In this study, we used TCGA datasets to demonstrate that TRIM59 was upregulated in multiple cancers in comparison to the adjacent normal tissues. To explore the diagnosis efficacy of TRIM59, we performed ROC and SROC analyses to identify that high expression of TRIM59 showed a high diagnosis efficacy in each tumor type (AUC > 0.5), and with an AUC of summary ROC of 0.94 (0.91–0.96). We also combined GEO datasets and performed a meta-analysis to reveal that high expression of TRIM59 showed significant (p<0.05) poor prognosis in KIRP, LGG, LUAD, lung cancer and showed better prognosis in CESC and SKCM. Especially

**Table 2. Characteristics of studies in OS analysis.**

| Study | Country | Year | Patients | HR | Lower | Upper |
|---|---|---|---|---|---|---|
| TCGA (BLCA) | USA | 2014 | 402 | 0.86 | 0.64 | 1.16 |
| TCGA (BRCA) | USA | 2014 | 1006 | 1.25 | 0.89 | 1.75 |
| TCGA (CESC) | USA | 2014 | 264 | 0.56 | 0.34 | 0.92 |
| TCGA (COAD) | USA | 2014 | 440 | 1.23 | 0.83 | 1.83 |
| TCGA (ESCA) | USA | 2014 | 144 | 1.17 | 0.70 | 1.97 |
| TCGA (GBM) | USA | 2014 | 152 | 1.07 | 0.75 | 1.53 |
| TCGA (HNSC) | USA | 2014 | 496 | 1.26 | 0.97 | 1.66 |
| TCGA (KIRC) | USA | 2014 | 522 | 1.28 | 0.95 | 1.73 |
| TCGA (KIRP) | USA | 2014 | 284 | 3.18 | 1.76 | 5.75 |
| TCGA (LGG) | USA | 2014 | 510 | 1.98 | 1.39 | 2.83 |
| TCGA (LIHC) | USA | 2014 | 360 | 1.41 | 0.99 | 2.00 |
| TCGA (LUAD) | USA | 2014 | 492 | 1.40 | 1.04 | 1.88 |
| TCGA (LUSC) | USA | 2014 | 488 | 0.84 | 0.64 | 1.10 |
| TCGA (OV) | USA | 2014 | 294 | 1.03 | 0.77 | 1.38 |
| TCGA (PAAD) | USA | 2014 | 174 | 1.46 | 0.97 | 2.20 |
| TCGA (READ) | USA | 2014 | 158 | 1.50 | 0.68 | 3.28 |
| TCGA (SARC) | USA | 2014 | 258 | 0.93 | 0.63 | 1.38 |
| TCGA (SKCM) | USA | 2014 | 458 | 0.71 | 0.54 | 0.92 |
| TCGA (STAD) | USA | 2014 | 378 | 0.81 | 0.59 | 1.12 |
| TCGA (UCEC) | USA | 2014 | 540 | 1.22 | 0.80 | 1.84 |
| GSE20685 (Breast Cancer) | China | 2011 | 327 | 1.31 | 0.85 | 2.01 |
| GSE20711 (Breast Cancer) | Canada | 2011 | 88 | 1.23 | 0.56 | 2.68 |
| GSE16446 (Breast Cancer) | Canada | 2010 | 107 | 1.28 | 0.46 | 3.54 |
| GSE42568 (Breast Cancer) | Ireland | 2013 | 104 | 1.02 | 0.53 | 1.99 |
| GSE48390 (Breast Cancer) | China | 2014 | 81 | 0.86 | 0.26 | 2.79 |
| GSE58812 (Breast Cancer) | France | 2015 | 107 | 0.77 | 0.37 | 1.60 |
| GSE102287 (Lung Cancer) | USA | 2017 | 66 | 1.20 | 0.64 | 2.23 |
| GSE29013 (Lung Cancer) | USA | 2011 | 55 | 1.31 | 0.52 | 3.30 |
| GSE30219 (Lung Cancer) | France | 2013 | 293 | 1.44 | 1.09 | 1.89 |
| GSE31210 (Lung Cancer) | Japan | 2011 | 226 | 2.37 | 1.22 | 4.60 |
| GSE3141 (Lung Cancer) | USA | 2005 | 111 | 0.96 | 0.57 | 1.60 |
| GSE19188 (Lung Cancer) | Netherlands | 2010 | 82 | 1.46 | 0.84 | 2.54 |
| GSE37745 (Lung Cancer) | Sweden | 2012 | 196 | 1.14 | 0.82 | 1.58 |
| GSE50081 (Lung Cancer) | Canada | 2013 | 181 | 1.21 | 0.77 | 1.90 |
| GSE26193 (Ovarian cancer) | France | 2011 | 107 | 0.78 | 0.50 | 1.23 |
| GSE32062 (Ovarian cancer) | Japan | 2012 | 260 | 1.03 | 0.72 | 1.48 |
| GSE63885 (Ovarian Cancer) | Poland | 2014 | 75 | 0.85 | 0.52 | 1.37 |
| GSE18520 (Ovarian adenocarcinomas) | USA | 2009 | 53 | 1.07 | 0.58 | 1.98 |
| GSE15459 (Gastric Cancer) | Switzerland | 2009 | 192 | 0.89 | 0.59 | 1.33 |
| GSE57303 (Gastric Cancer) | China | 2014 | 34 | 0.93 | 0.48 | 1.79 |
| GSE62254 (Gastric Cancer) | USA | 2015 | 295 | 0.78 | 0.57 | 1.08 |
| GSE17891 (Pancreatic Cancer) | United Kingdom | 2011 | 21 | 0.28 | 0.05 | 1.62 |
| GSE17679 (Ewing Sarcoma) | Finland | 2011 | 88 | 1.56 | 0.90 | 2.70 |
| GSE19750 (Adrenocortical carcinoma) | USA | 2013 | 22 | 0.39 | 0.15 | 1.05 |
| GSE108474 (Brain Cancer) | USA | 2018 | 487 | 0.96 | 0.78 | 1.19 |
| GSE7696 (Glioblastoma) | Switzerland | 2008 | 80 | 1.08 | 0.66 | 1.75 |

BLCA, bladder carcinoma; BRCA, breast carcinoma; CESC, cervical squamous cell carcinoma and endocervical adenocarcinoma; COAD, colon adenocarcinoma; ESCA, esophageal carcinoma; GBM, glioblastoma multiforme; HNSC, head and neck squamous cell carcinoma; KIRC, kidney renal clear cell carcinoma; KIRP, kidney renal papillary cell carcinoma; LGG, lower grade glioma; LIHC, liver hepatocellular carcinoma; LUAD, lung adenocarcinoma; LUSC, lung squamous cell carcinoma; OV, ovarian serous cystadenocarcinoma; PAAD, pancreatic adenocarcinoma; READ, rectum adenocarcinoma; SARC, sarcoma; SKCM, skin cutaneous melanoma; STAD, stomach adenocarcinoma; UCEC, uterine corpus endometrial carcinoma.

| Study | TE | seTE | Hazard Ratio | HR | 95%−CI | Weight (fixed) | Weight (random) |
|---|---|---|---|---|---|---|---|
| TCGA(BLCA) | −0.15 | 0.1513 | | 0.86 | [0.64; 1.16] | 3.9% | 3.2% |
| TCGA(BRCA) | 0.22 | 0.1722 | | 1.25 | [0.89; 1.75] | 3.0% | 2.9% |
| TCGA(CESC) | −0.59 | 0.2576 | | 0.56 | [0.34; 0.92] | 1.4% | 1.9% |
| TCGA(COAD) | 0.21 | 0.2021 | | 1.23 | [0.83; 1.83] | 2.2% | 2.5% |
| TCGA(ESCA) | 0.16 | 0.2655 | | 1.17 | [0.70; 1.97] | 1.3% | 1.9% |
| TCGA(GBM) | 0.07 | 0.1826 | | 1.07 | [0.75; 1.53] | 2.7% | 2.8% |
| TCGA(HNSC) | 0.23 | 0.1374 | | 1.26 | [0.97; 1.66] | 4.8% | 3.4% |
| TCGA(KIRC) | 0.25 | 0.1521 | | 1.28 | [0.95; 1.73] | 3.9% | 3.2% |
| TCGA(KIRP) | 1.16 | 0.3016 | | 3.18 | [1.76; 5.75] | 1.0% | 1.6% |
| TCGA(LGG) | 0.68 | 0.1816 | | 1.98 | [1.39; 2.83] | 2.7% | 2.8% |
| TCGA(LIHC) | 0.34 | 0.1787 | | 1.41 | [0.99; 2.00] | 2.8% | 2.8% |
| TCGA(LUAD) | 0.34 | 0.1499 | | 1.40 | [1.04; 1.88] | 4.0% | 3.2% |
| TCGA(LUSC) | −0.17 | 0.1375 | | 0.84 | [0.64; 1.10] | 4.8% | 3.4% |
| TCGA(OV) | 0.03 | 0.1499 | | 1.03 | [0.77; 1.38] | 4.0% | 3.2% |
| TCGA(PAAD) | 0.38 | 0.2096 | | 1.46 | [0.97; 2.20] | 2.1% | 2.4% |
| TCGA(READ) | 0.40 | 0.4000 | | 1.50 | [0.68; 3.28] | 0.6% | 1.0% |
| TCGA(SARC) | −0.07 | 0.2020 | | 0.93 | [0.63; 1.38] | 2.2% | 2.5% |
| TCGA(SKCM) | −0.35 | 0.1364 | | 0.71 | [0.54; 0.92] | 4.8% | 3.4% |
| TCGA(STAD) | −0.21 | 0.1628 | | 0.81 | [0.59; 1.12] | 3.4% | 3.0% |
| TCGA(UCEC) | 0.20 | 0.2108 | | 1.22 | [0.80; 1.84] | 2.0% | 2.4% |
| GSE20685(Breast Cancer) | 0.27 | 0.2195 | | 1.31 | [0.85; 2.01] | 1.9% | 2.3% |
| GSE20711(Breast Cancer) | 0.20 | 0.4001 | | 1.23 | [0.56; 2.68] | 0.6% | 1.0% |
| GSE16446(Breast Cancer) | 0.25 | 0.5180 | | 1.28 | [0.46; 3.54] | 0.3% | 0.7% |
| GSE42568(Breast Cancer) | 0.02 | 0.3381 | | 1.02 | [0.53; 1.99] | 0.8% | 1.3% |
| GSE48390(Breast Cancer) | −0.16 | 0.6031 | | 0.86 | [0.26; 2.79] | 0.2% | 0.5% |
| GSE58812(Breast Cancer) | −0.26 | 0.3715 | | 0.77 | [0.37; 1.60] | 0.7% | 1.2% |
| GSE102287(Lung Cancer) | 0.18 | 0.3165 | | 1.20 | [0.64; 2.23] | 0.9% | 1.5% |
| GSE29013(Lung Cancer) | 0.27 | 0.4715 | | 1.31 | [0.52; 3.30] | 0.4% | 0.8% |
| GSE30219(Lung Cancer) | 0.36 | 0.1416 | | 1.44 | [1.09; 1.89] | 4.5% | 3.3% |
| GSE31210(Lung Cancer) | 0.86 | 0.3383 | | 2.37 | [1.22; 4.60] | 0.8% | 1.3% |
| GSE3141(Lung Cancer) | −0.04 | 0.2627 | | 0.96 | [0.57; 1.60] | 1.3% | 1.9% |
| GSE19188(Lung Cancer) | 0.38 | 0.2835 | | 1.46 | [0.84; 2.54] | 1.1% | 1.7% |
| GSE37745(Lung Cancer) | 0.13 | 0.1662 | | 1.14 | [0.82; 1.58] | 3.3% | 3.0% |
| GSE50081(Lung Cancer) | 0.19 | 0.2309 | | 1.21 | [0.77; 1.90] | 1.7% | 2.2% |
| GSE26193(Ovarian cancer) | −0.24 | 0.2296 | | 0.78 | [0.50; 1.23] | 1.7% | 2.2% |
| GSE32062(Ovarian cancer) | 0.03 | 0.1818 | | 1.03 | [0.72; 1.48] | 2.7% | 2.8% |
| GSE63885(Ovarian Cancer) | −0.17 | 0.2462 | | 0.85 | [0.52; 1.37] | 1.5% | 2.0% |
| GSE18520(Ovarian adenocarcinomas) | 0.07 | 0.3129 | | 1.07 | [0.58; 1.98] | 0.9% | 1.5% |
| GSE15459(Gastric Cancer) | −0.12 | 0.2055 | | 0.89 | [0.59; 1.33] | 2.1% | 2.5% |
| GSE57303(Gastric Cancer) | −0.07 | 0.3334 | | 0.93 | [0.48; 1.79] | 0.8% | 1.4% |
| GSE62254(Gastric Cancer) | −0.25 | 0.1634 | | 0.78 | [0.57; 1.08] | 3.4% | 3.0% |
| GSE17891(Pancreatic Cancer) | −1.27 | 0.8959 | | 0.28 | [0.05; 1.62] | 0.1% | 0.3% |
| GSE17679(Ewing Sarcoma) | 0.45 | 0.2789 | | 1.56 | [0.90; 2.70] | 1.2% | 1.7% |
| GSE19750(Adrenocortical carcinoma) | −0.94 | 0.5020 | | 0.39 | [0.15; 1.05] | 0.4% | 0.7% |
| GSE108474(Brain Cancer) | −0.04 | 0.1077 | | 0.96 | [0.78; 1.19] | 7.8% | 3.8% |
| GSE7696(Glioblastoma) | 0.07 | 0.2484 | | 1.08 | [0.66; 1.75] | 1.5% | 2.0% |
| **Fixed effect model** | | | | **1.08** | **[1.02; 1.15]** | **100.0%** | **−−** |
| **Random effects model** | | | | **1.10** | **[1.01; 1.20]** | **−−** | **100.0%** |

Heterogeneity: $I^2$ = 51%, $\tau^2$ = 0.0439, $p$ < 0.01

0.1   0.5  1  2     10

**Fig 4. Forest plot showing the overall survival significance of TRIM59.** Patients were divided into high expression and low expression groups based on the median expression levels of TRIM59, then overall survival was compared between the high expression and low expression group.

in LUAD, the expression of TRIM59 in tumors with better prognosis was still higher than that in adjacent tissues (S1 Fig).

TRIM59 is closely related to cancers. A previous study used Immunohistochemistry (IHC) to determine the expression of TRIM59 in 291 cases of 37 tumor types, and found that TRIM59 expression was upregulated in tumor samples, particularly in lung, breast, liver, skin, tongue and mouth (squamous cell cancer) and endometrial cancers [9]. In subsequent studies, it was shown that upregulation of TRIM59 can promote tumor growth in tumor cell lines and animal models, while downregulation had the opposite effect. These cancers include pancreatic cancer [13], cholangiocarcinoma [14], ovarian cancer [15, 16], lung cancer [17–19], breast cancer [10, 20, 21], euroblastoma [22], medulloblastoma [23], hepatocellular carcinoma [24], glioblastoma [11], colorectal cancer [25, 26], bladder cancer [21], prostate cancer [27], cervical cancer [28], osteosarcoma [29], gastric cancer [30]. However, results from these studies were limited by a small number of tumor samples, or the use of in *vitro* experimentation or animal models could not fully address the relationship between TRIM59 and human cancers. Thanks to the gene expression data and associated prognosis information provided in public database, we identified that TRIM59 was highly expressed in most solid tumors and could indicate the prognosis in several cancers.

This study is a meta-analysis of multiple solid tumors, indicated that TRIM59 has the potential to be used as a diagnostic molecule for a variety of tumors, and special attention should be paid to the abnormal high expression of TRIM59 in specific tissues. Moreover, it plays a prognostic role in specific tumors, especially in KIRP/LGG/LUAD/Lung cancer/CESC/SKCM. Detection of TRIM59 expression in these tumor tissues is helpful for us to evaluate the prognosis of patients.

The limitation of this study lies in the fact that it is only the data analysis of biological database, and further verification of TRIM59 expression level and follow-up information are needed in clinical samples for each specific tumor type. Basic research on TRIM59 is also needed to improve our understanding of tumors.

## Conclusions

In this study, based on TCGA datasets, we revealed that TRIM59 was upregulated in 15 types of solid tumors. Additionally, TRIM59 have a high efficacy in diagnosis and prognosis prediction in various tumor types. TRIM59 have the potential to be used as diagnosis marker or prognosis predictor in tumors.

## Supporting information

**S1 Fig. Expression of TRIM59 in LUAD-low expression and KIRP-low expression groups and associated adjacent tissues.**
(TIF)

## Author Contributions

**Conceptualization:** Dongmei Yan.

**Data curation:** Zheng Jin, Youran Yu, Zhenhua Zhu.

**Formal analysis:** Xun Zhu.

**Funding acquisition:** Dongmei Yan.

**Investigation:** Zheng Jin.

**Methodology:** Liping Liu, Zhenhua Zhu.

**Software:** Youran Yu, Dong Li.

**Supervision:** Xun Zhu, Dongmei Yan.

**Validation:** Liping Liu, Dong Li, Zhenhua Zhu.

**Visualization:** Dong Li.

**Writing – original draft:** Zheng Jin.

**Writing – review & editing:** Xun Zhu, Dongmei Yan, Zhenhua Zhu.

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
