## [Decision Letter · Decision Letter 0]

4 Mar 2021

PONE-D-20-32431

TRIM59: a potential diagnostic and prognostic biomarker in human tumors

PLOS ONE

Dear Dr. Zhenhua Zhu,

Thank you for submitting your manuscript to PLOS ONE. After careful consideration, we feel that it has merit but does not fully meet PLOS ONE’s publication criteria as it currently stands. Therefore, we invite you to submit a revised version of the manuscript that addresses the points raised during the review process.

We look forward to receiving your revised manuscript.

Kind regards,

Suhwan Chang

Academic Editor

PLOS ONE

Additional Editor Comments:

I decide to proceed with the one reviewer's comment. Please check it carefully and upload revised version.

Thanks

Journal Requirements:

2. In the Methods section, please provide the accession numbers of the datasets downloaded from GEO for your study.

3. To comply with PLOS ONE submission guidelines, in your Methods section, please provide additional information regarding your statistical analyses. For more information on PLOS ONE's expectations for statistical reporting, please see https://journals.plos.org/plosone/s/submission-guidelines.#loc-statistical-reporting.

Reviewers' comments:

Reviewer's Responses to Questions

**Comments to the Author**

1. Is the manuscript technically sound, and do the data support the conclusions?

Reviewer #1: Partly

2. Has the statistical analysis been performed appropriately and rigorously? 

Reviewer #1: I Don't Know

3. Have the authors made all data underlying the findings in their manuscript fully available?

Reviewer #1: Yes

4. Is the manuscript presented in an intelligible fashion and written in standard English?

Reviewer #1: Yes

5. Review Comments to the Author

Reviewer #1: This mansucript reports a meta-analysis of TRIM-59 expression in a number of large publicly-available cancer datasets, particularly TCGA and GEO. The results show that TRIM-59 expression is consistently higher in tumor tissue relative to its matching benign tissue in each study, however it is expressed in both tissue types. In addition, expression level was looked at as a prognostic marker and found to be higher in cases with worse outcome for a number of tumor types, but not all.

The strengths of the study are the large number of cases examined from reliable data sources. Also analyses were performed to evaluate for biases typical of these types of study designs.

The main weakness of the study is the claim that the data shows this to be a good diagnostic biomarker for cancer, however the study was not designed to evaluate for this being a diagnostic biomarker. IN fact, the data seem to indicate that this is a very poor diagnostic biomarker, as figure 1 shows many benign tissue types to have higher TRIM-59 levels than some tumor types. Also, there is an opportunity to look at great detail for certain tumor types but the opportunity was not taken.

Specific areas to consider:

1. figure 1, what does FPKM stand for? should indicate in the figure itself or in the figure caption.

2. All claims about this being a good diagnostic biomarker should be excluded, as none of this data shows such. In fact, the data actually could be interpreted that it shows the opposite at this point in time (some benign levels are higher than some cancer levels and the ROC is not good enough for diagnostic purposes). This study was not designed to determine if this is a diagnostic biomarker or not. From this data, ONe can conclude that TRIM-59 is worth examining as a diagnostic biomarker (and that can be stated here), but studies need to be properly designed to evaluate for this.

3. In realtion to point 2, how did the expression levels in benign compare to the good outcome cancers for each tumor type??? This was not included, but should be looked at to further see if this is still a good diagnostic biomarker when that comparison is done.

4. In the discussion, maybe more about the limitations of this study, including that such meta-analyses are evaluating large datasets of collections of many tumor types, and that clinical utility would need to be looked at for each different tumor type in relation to clinical needs rather than the entire collection as a whole.

6. PLOS authors have the option to publish the peer review history of their article (what does this mean?). If published, this will include your full peer review and any attached files.

Reviewer #1: No

---

## [Author Response · Author response to Decision Letter 0]

22 Mar 2021

Answer: We have modified the manuscript according to the style templates as required.

2. In the Methods section, please provide the accession numbers of the datasets downloaded from GEO for your study.

Answer: We have provided the accession numbers as required. (line 66-74)

3. To comply with PLOS ONE submission guidelines, in your Methods section, please provide additional information regarding your statistical analyses. For more information on PLOS ONE's expectations for statistical reporting, please see https://journals.plos.org/plosone/s/submission-guidelines.#loc-statistical-reporting.

Answer: We have added additional information regarding statistical analyses in the method section as required. (line 76-95)

Review Comments to the Author

01

Reviewer #1: This mansucript reports a meta-analysis of TRIM-59 expression in a number of large publicly-available cancer datasets, particularly TCGA and GEO. The results show that TRIM-59 expression is consistently higher in tumor tissue relative to its matching benign tissue in each study, however it is expressed in both tissue types. In addition, expression level was looked at as a prognostic marker and found to be higher in cases with worse outcome for a number of tumor types, but not all.

The strengths of the study are the large number of cases examined from reliable data sources. Also analyses were performed to evaluate for biases typical of these types of study designs.

The main weakness of the study is the claim that the data shows this to be a good diagnostic biomarker for cancer, however the study was not designed to evaluate for this being a diagnostic biomarker. IN fact, the data seem to indicate that this is a very poor diagnostic biomarker, as figure 1 shows many benign tissue types to have higher TRIM-59 levels than some tumor types. Also, there is an opportunity to look at great detail for certain tumor types but the opportunity was not taken.

Answer: Thank you for your genius opinions. What we should first explain is that in the previous data processing of TCGA datasets, UCSC pan-tumor mean-normalized gene expression data was used. In the process of normalization, gene values are mean-centered per gene. This is useful for correlation analysis between different tumor types, but it weakens the differences in actual expression levels between samples. Therefore, in present analysis, we used the original FPKM expression data of each sample for analysis (line 89-91). As shown in figure 1, all the tumors showed significant higher expression of TRIM59 than their corresponding adjacent tissues. Moreover, comparisons between tumor types are not very useful due to differences in tumor material procession and batch effects. In clinical practice, we only need to consider the amount of TRIM59 expression in the suspected tumor tissue and the adjacent normal tissue. The purpose of this study was to investigate the feasibility of TRIM59 as a diagnostic and prognostic molecule for a variety of tumors.

Specific areas to consider:

figure 1, what does FPKM stand for? should indicate in the figure itself or in the figure caption.

Answer: Thank you for your kind reminding. FPKM (Fragments Per Kilobase of exon model per Million mapped fragments) , FPKM=(total exon Fragments)/(mapped reads(Millions)×exon length(KB)) . In transcriptome sequencing data, FPKM reflects the expression level of genes. We have indicated it in figure caption. (line 332)

All claims about this being a good diagnostic biomarker should be excluded, as none of this data shows such. In fact, the data actually could be interpreted that it shows the opposite at this point in time (some benign levels are higher than some cancer levels and the ROC is not good enough for diagnostic purposes). This study was not designed to determine if this is a diagnostic biomarker or not. From this data, ONe can conclude that TRIM-59 is worth examining as a diagnostic biomarker (and that can be stated here), but studies need to be properly designed to evaluate for this.

Answer: Thank you for your kind reminding. As shown in figure 1, tumor tissues showed significant higher expression than corresponding adjacent tissues in each tumor type. And by setting a threshold, the tumors can be well distinguished from adjacent tissues. For these reasons, as suggested by you, we concluded that TRIM59 is worth examining as a diagnostic biomarker in multiple tumors. 

In realtion to point 2, how did the expression levels in benign compare to the good outcome cancers for each tumor type??? This was not included, but should be looked at to further see if this is still a good diagnostic biomarker when that comparison is done.

Answer: We appreciate for your kind reminding. We agree that this comparison is indeed necessary. After comparison, it was found that the expression level of TRIM59 in LUAD with better prognosis was still significantly higher than that in normal adjacent tissues, however no significant difference was observed between good prognosis group and adjacent tissue in KIRP (S1 Fig, line 166-168). Because the datasets did not contain sufficient adjacent samples, such comparisons could not be made in LGG/CESC/SKCM.

In the discussion, maybe more about the limitations of this study, including that such meta-analyses are evaluating large datasets of collections of many tumor types, and that clinical utility would need to be looked at for each different tumor type in relation to clinical needs rather than the entire collection as a whole.

Answer: Thank you. We have made substantial modifications to the discussion section as suggested. And we did not discuss all tumors as a whole in the prognostic analysis in the present version of manuscript.

---

## [Editor Report · Decision Letter 1]

2 Sep 2021

TRIM59: a potential diagnostic and prognostic biomarker in human tumors

PONE-D-20-32431R1

Dear Dr. Zhu,

We’re pleased to inform you that your manuscript has been judged scientifically suitable for publication and will be formally accepted for publication once it meets all outstanding technical requirements.

Kind regards,

Salvatore V Pizzo

Academic Editor

PLOS ONE
---

## [Editor Report · Acceptance letter]

10 Sep 2021

PONE-D-20-32431R1 

TRIM59: a potential diagnostic and prognostic biomarker in human tumors 

Dear Dr. Zhu:

I'm pleased to inform you that your manuscript has been deemed suitable for publication in PLOS ONE. Congratulations! Your manuscript is now with our production department. 

Kind regards, 

on behalf of

Dr. Salvatore V Pizzo 

Academic Editor

PLOS ONE